# A histone H3K4me1-specific binding protein is required for siRNA accumulation and DNA methylation at a subset of loci targeted by RNA-directed DNA methylation

Qingfeng Niu [1,6], Zhe Song [1,2,6], Kai Tang [3], Lixian Chen[1,2], Lisi Wang[1,2], Ting Ban[1], Zhongxin Guo[4], Chanhong Kim [1], Heng Zhang[1], Cheng-Guo Duan [1], Huiming Zhang [1], Jian-Kang Zhu [1], Jiamu Du [5✉] & Zhaobo Lang [1✉]

In plants, RNA-directed DNA methylation (RdDM) is a well-known de novo DNA methylation pathway that involves two plant-specific RNA polymerases, Pol IV and Pol V. In this study, we discovered and characterized an RdDM factor, RDM15. Through DNA methylome and genome-wide siRNA analyses, we show that RDM15 is required for RdDM-dependent DNA methylation and siRNA accumulation at a subset of RdDM target loci. We show that RDM15 contributes to Pol V-dependent downstream siRNA accumulation and interacts with NRPE3B, a subunit specific to Pol V. We also show that the C-terminal tudor domain of RDM15 specifically recognizes the histone 3 lysine 4 monomethylation (H3K4me1) mark. Structure analysis of RDM15 in complex with the H3K4me1 peptide showed that the RDM15 tudor domain specifically recognizes the monomethyllysine through an aromatic cage and a specific hydrogen bonding network; this chemical feature-based recognition mechanism differs from all previously reported monomethyllysine recognition mechanisms. RDM15 and H3K4me1 have similar genome-wide distribution patterns at RDM15-dependent RdDM target loci, establishing a link between H3K4me1 and RDM15-mediated RdDM in vivo. In summary, we have identified and characterized a histone H3K4me1-specific binding protein as an RdDM component, and structural analysis of RDM15 revealed a chemical feature-based lower methyllysine recognition mechanism.

[1] Shanghai Center for Plant Stress Biology, National Key Laboratory of Plant Molecular Genetics, Center for Excellence in Molecular Plant Sciences, Chinese Academy of Sciences, Shanghai, China. [2] University of Chinese Academy of Sciences, Beijing, China. [3] Department of Horticulture and Landscape Architecture, Purdue University, West Lafayette, IN, USA. [4] Vector-borne Virus Research Center, College of Plant Protection, Fujian Agriculture and Forestry Universtiy, Fuzhou, China. [5] Key Laboratory of Molecular Design for Plant Cell Factory of Guangdong Higher Education Institutes, Institute of Plant and Food Science, School of Life Science, Southern University of Science and Technology, Shenzhen, Guangdong, China. [6] These authors contributed equally: Qingfeng Niu, Zhe Song. ✉email: dujm@sustech.edu.cn; zblang@psc.ac.cn

DNA methylation mainly refers to an addition of a methyl group to the fifth position of cytosine, resulting in 5′-methylcytosine (5-mC). Such methylation is involved in gene imprinting, fruit ripening, maintenance of genome integrity, and other important biological functions[1]. In mammals, DNA methylation mainly occurs in the CG context, but can be found in all three sequence contexts, including CG, CHG, and CHH (H = T, A, or C) in plants[2]. Cytosines in different sequence contexts can be methylated by different methyltransferases: cytosine in all sequence contexts can be de novo methylated by DRM2 (DOMAINS REARRANGED METHYLTRANSFERASE 2) through the RNA-dependent DNA methylation (RdDM) pathway; CG and CHG methylation is maintained by MET1 (METHYLTRANSFERASE 1) and CMT3 (CHROMOMETHY-LASE 3), respectively, while CHH methylation is maintained by CMT2 or DRM2, depending on the chromatin context[1,3–8].

The RdDM pathway is responsible for DNA methylation and transcriptional silencing of transposons and other repetitive elements and is critical for pathogen defense, stress response, and many other processes in plants[1,9,10]. This pathway comprises two main steps that depend on the plant-specific RNA polymerases Pol IV and Pol V. The first step is Pol IV-dependent siRNA biogenesis that involves not only Pol IV but also several other RdDM factors, such as RDR2 (RNA-DEPENDENT RNA POLYMERASE 2) and DCL3 (DICER-LIKE 3). The second step is siRNA-guided DNA methylation, in which Pol V-transcribed long-noncoding RNA serves as a scaffold RNA, and DRM2 is recruited to catalyze DNA methylation. This recruitment of DRM2 involves base-pairing between AGO4 (ARGONAUTE 4)- and AGO6-bound siRNAs and the scaffold RNA and the participation of RdDM factors such as KTF1 (KOW DOMAIN-CONTAINING TRANSCRIPTION FACTOR 1), DRD1, DMS3, and RDM1 (RNA-DIRECTED DNA METHYLATION 1)[10,11].

Different epigenetic marks, such as DNA methylation and histone modifications, can interact to regulate the chromatin state and DNA methylation level in the genome[12]. Histone modifications are established by histone modification enzymes known as writer proteins and can be recognized by reader proteins to regulate downstream effector molecules. To date, with few exceptions, all of the reported recognition mechanisms of histone mark readers are conserved between plants and animals[13,14].

In RdDM, the recruitment of Pol IV requires SHH1 (SAWA-DEE HOMEODOMAIN HOMOLOG 1), which possesses an SAWADEE domain that can bind to both unmethylated H3K4 and methylated H3K9[15,16]. At RdDM target loci, active histone marks, such as H3K4me2 and H3K4me3, function to prevent RdDM[17]. On the other hand, the recruitment of Pol V requires the participation of SUVH2 and SUVH9[18,19]. Although these two proteins lack histone methyltransferase activities, they both have an SRA domain that can bind to methylated DNA[19,20], indicating that they contribute to the interplay between Pol V and chromatin by binding to pre-existing methylated DNA. The crosstalk between histone modification and RdDM has been extensively studied, but the existing model cannot explain all of the targeting mechanisms of DNA methylation. Additional studies are required to increase our understanding of the interaction between epigenetic modification and RdDM components.

In the current research, we found a chromatin-binding protein, RDM15, that is involved in RdDM. Through whole-genome bisulfite sequencing and siRNA sequencing of rdm15 mutants and the wild type (WT), we demonstrate that RDM15 is required for RdDM-dependent DNA methylation and RdDM-dependent siRNA accumulation at a subset of RdDM target regions. In addition, protein structure analysis revealed that RDM15 can specifically recognize H3K4me1 through its C-terminal Tudor domain. The RDM15 Tudor domain uses an aromatic cage and a

hydrogen-bonding network to achieve the monomethyllysine-specific recognition of H3K4me1; this chemical feature-based monomethylation state-specific readout mechanism has not been discovered previously. Our results also show RDM15 protein enrichment at RDM15-dependent RdDM target regions and physical interaction between RDM15 and a PolV subunit. In summary, our study has identified an RdDM component, RDM15, and has elucidated the molecular mechanism underlying RDM15 function in the RdDM pathway.

## Results

**RDM15 is required for transcriptional silencing at RdDM target loci.** In this study, we used a T-DNA mutagenized population in the Arabidopsis ros1-1 mutant background (C24 ecotype) to identify genetic factors involved in DNA methylation and transcriptional gene silencing. This screen was based on a stress-responsive RD29A promoter-driven LUCIFERASE transgene (pRD29A-LUC) in the ros1-1 mutant background. Dysfunction of ROS1 silences the pRD29A-LUC transgene and endogenous RD29A gene through heavy methylation at the RD29A promoter[21]. A CaMV 35S promoter-driven kanamycin-resistance gene NPTII (p35S-NPTII) that is linked to the pRD29A-LUC transgene is also silenced in the ros1-1 mutant[21]. By screening for ros1 suppressor mutants, we have discovered several components of the RdDM pathway, such as RDM1, RDM4, and KTF1[22,23]. In previous studies, all RdDM mutants isolated through this screen can release the silencing of pRD29A-LUC but not of p35S-NPTII, because silencing of p35S-NPTII is not fully dependent on the RdDM pathway[22,23]. In the current study, we isolated a ros1-1 suppressor mutant, rdm15-1, that can partially release transcriptional silencing of pRD29A-LUC in the ros1-1 background but cannot release the silencing of p35S-NPTII (Fig. 1a and Supplementary Fig. 1a). As shown in Fig. 1a, the luminescence after cold treatment was partially recovered in the rdm15-1/ros1-1 double mutant compared to ros1-1. Consistently, the transcript level of LUC was higher in the rdm15-1/ros1-1 double mutant than in ros1-1 (Fig. 1b). The silencing of the endogenous RD29A gene in ros1-1 was also released by the rdm15 mutation (Fig. 1b). However, the silencing of p35S-NPTII in ros1-1 was not released by the rdm15-1 mutation (Supplementary Fig. 1a). In addition to silencing the transgene, RDM15 is also required for the silencing of endogenous genomic loci. AtGP1 and MEA-ISR are known RdDM targets that are silenced in WT and ros1-1 mutant plants, and this silencing can be released in ago4, a known RdDM mutant, and also in rdm15 mutants (Supplementary Fig. 1b)[1]. These results suggest that RDM15 is involved in RdDM-dependent gene silencing.

To clone the RDM15 gene, we used TAIL-PCR and found a T-DNA insertion in the 7th exon of AT4G31880 (Fig. 1c). A complementation test showed that a transgene containing the native promoter-driven genomic sequence of AT4G31880 can rescue the pRD29A-LUC silencing phenotype of the rdm15-1/ros1-1 double mutant (Supplementary Fig. 1c). To confirm the silencing function of RDM15, we ordered two T-DNA insertion alleles of RDM15 (Col-0 background): Salk_013481 and Salk_024055 (hereafter referred to as rdm15-2 and rdm15-3). Both of these mutants contain T-DNA insertions in the 7th exon of RDM15 (Fig. 1c and Supplementary Fig. 1d). The silencing of AtGP1 and AtMEA-ISR, two known RdDM targets[23], is released by both rdm15-2 and rdm15-3 mutations (Fig. 1d), supporting the notion that RDM15 is required for the silencing of RdDM target loci.

RdDM controls the expression of the DNA demethylase ROS1 via methylation of the MEMS sequence located in the promoter region of ROS1[24,25]. ROS1 expression is dramatically decreased in

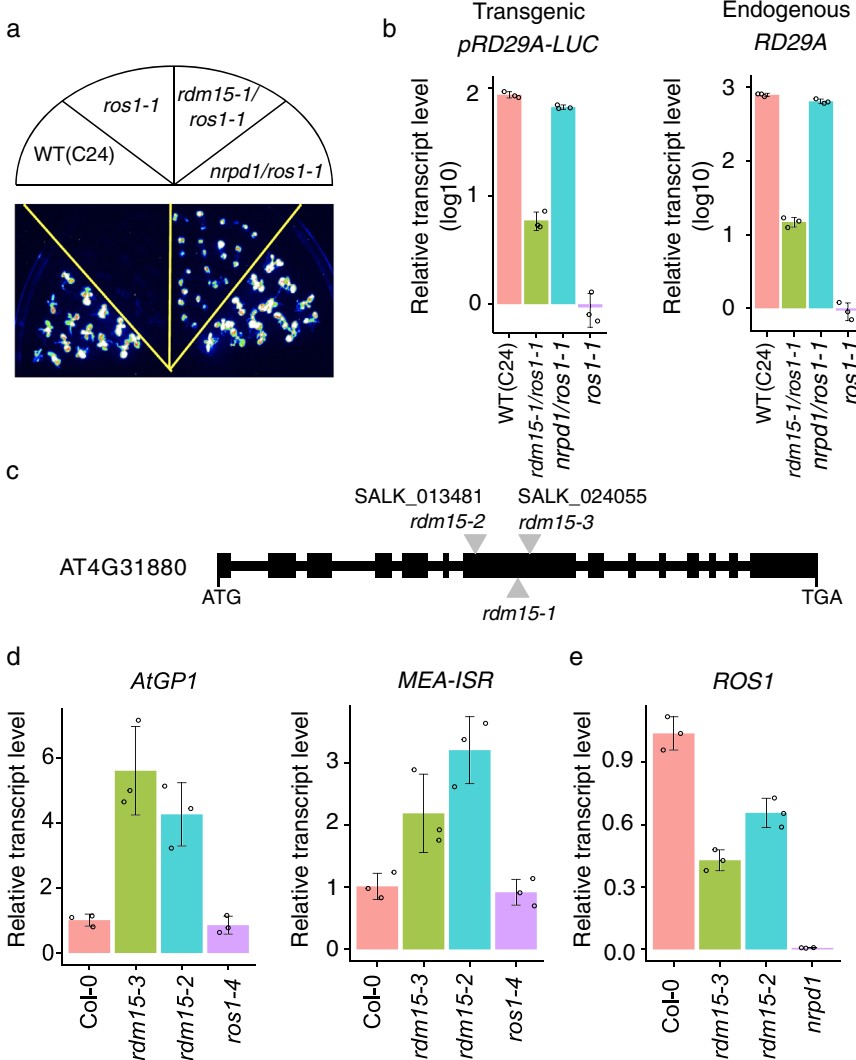

**Fig. 1 RDM15 mediates transcriptional silencing. a** Isolation of the *rdm15-1* mutant. The wild-type (WT) of the C24 ecotype carries a stress-inducible *RD29A-LUC* transgene, the expression of which can be assessed by luminescence. The WT, *ros1-1*, *rdm15-1ros1-1*, and *nrpd1ros1-1* plants were grown for 10 days and imaged after cold treatment (48 h, 4 °C). **b** RT-qPCR analysis of relative transcript levels of transgenic *RD29A-LUC* and endogenous *RD29A* in the WT, *rdm15-1 ros1-1*, *nrpd1 ros1-1*, and *ros1-1*. Ten-day-old seedlings were used for RNA extraction after cold treatment (48 h, 4 °C). **c** Diagram showing the positions of T-DNA insertions in *rdm15-1*, *rdm15-2*, and *rdm15-3*. Boxes and lines denote exons and introns of *RDM15* (AT4G31880), respectively. **d** RT-qPCR analysis of relative transcript levels of *AtGP1* and *MEA_ISR* in Col−0 (WT), *rdm15-2*, *rdm15-3*, and *ros1-4*. **e** RT-qPCR analysis of relative transcript level of *ROS1* in Col-0, *rdm15-2*, *rdm15-3*, and *nrpd1-3*. *ACTIN2* served as the internal control in the RT-qPCR analysis. Error bars represent s.d. ($n = 3$ biologically independent samples).

many RdDM mutants, including *nrpd1a*, *nrpd1b*, and *ago4*[24,25]. We found that *rdm15* mutants have lower *ROS1* transcript levels (Fig. 1e and Supplementary Fig. 1e), although the levels are not as low as in *nrpd* or *ago4* mutants. These data suggest that RDM15 functions in RdDM-dependent transcriptional gene silencing (TGS).

**RDM15 is required for RdDM-dependent DNA methylation.** To characterize the effect of *rdm15* on DNA methylation, we generated single-base resolution maps of DNA methylation for Col-0, *rdm15-2*, and *rdm15-3*, with two biological replicates for each mutant. Principal components analysis showed very good consistency between the replicates (Supplementary Fig. 2a). Using the R package methylKit[26], which considers variations among replicates of two mutant alleles, we identified 1390 differentially methylated regions (DMRs) in *rdm15* mutants compared with the WT; most of the DMRs (1354 out of 1390) were hypomethylated,

suggesting that RDM15 is required for DNA methylation at more than 1300 endogenous genomic regions (Supplementary Data 1). We compared the DNA methylation patterns of *rdm15* mutants with those of known RdDM mutants. NRPD1 and NRPE1 are the largest subunits exclusive to Pol IV and Pol V, respectively. Using the same methylKit method and compared to the WT, we identified 4293 hypo DMRs in *nrpd1-3* (Col-0 ecotype) and 4629 hypo DMRs in *nrpe1-11* (Col-0 ecotype) (Supplementary Data 1). We found that the Pol IV, Pol V, and RDM15 targets have comparable genomic compositions. Among the hypo DMRs, 82% in *nrpd1–3*, 81% in *nrpe1–11*, and 79% in *rdm15* are in TE regions (Fig. 2a). Similar to known RdDM targets, *rdm15* DMRs show DNA hypomethylation in all three sequence contexts (mCG, mCHG, and mCHH) compared to WT (Fig. 2b), although the methylation level in *rdm15* mutants is not as low as in *nrpd1–3* (Fig. 2b). We further analyzed the overlap between RDM15 targets and known RdDM targets. As shown in Fig. 2c,

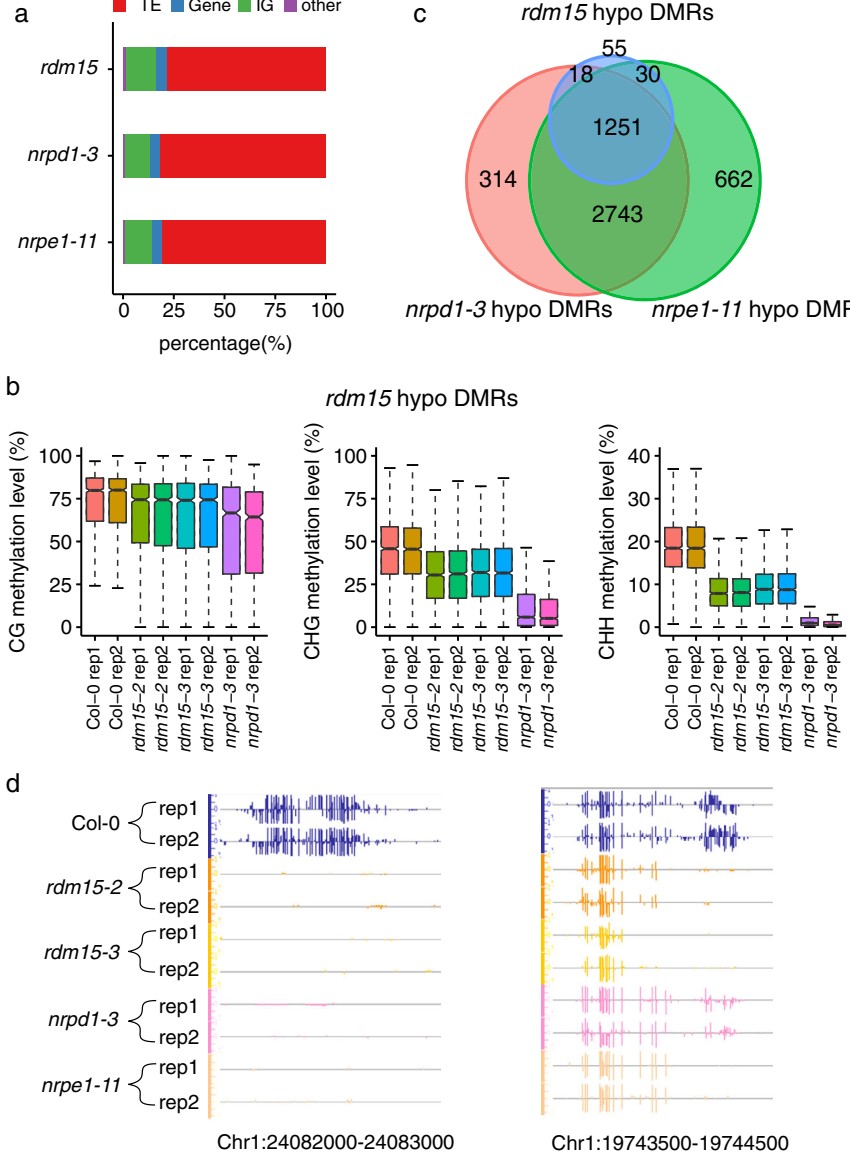

**Fig. 2 RDM15 is required for RdDM-dependent DNA methylation. a** Compositions of DMRs among TEs, genes, and intergenic regions (IGs). DMRs (relative to the WT) were identified in *rdm15*, *nrpd1-3*, and *nrpe1-11*. **b** Box plots showing CG, CHG, and CHH methylation levels of *rdm15* DMRs in Col-0, *rdm15-2*, *rdm15-3*, and *nrpd1-3*. Data from two replicates are shown. **c** Venn diagram showing the numbers of hypomethylated DMRs that overlapped between *rdm15*, *nrpd1-3*, and *nrpe1-11*. **d** The Integrative Genome Browser (IGB) display of whole-genome bisulfite sequencing (WGBS) data is shown for several representative *rdm15* DMRs in Col-0, *rdm15-2*, *rdm15-3*, and *nrpd1-3*. Vertical bars on each track indicate DNA methylation levels. The coordinates of DMRs are indicated below the snapshots. Two replicates are shown for each genotype.

94% (1269/1354) and 95% (1281/1354) of *rdm15* hypo DMRs overlap with the *nrpd1* and *nrpe1* hypo DMRs, respectively. DNA methylation profiles of several representative *rdm15* hypo DMRs are shown in Fig. 2d and Supplementary Fig. 2b. These results revealed that RDM15 is required for methylation at a subset of RdDM target loci, suggesting that RDM15 may be an RdDM component.

**RDM15 regulates RdDM-dependent siRNA accumulation.** We generated genome-wide siRNA profiles for WT (Col−0), *rdm15-2*, and *rdm15-3* with two biological replicates for each genotype (Supplementary Fig. 3a). In *Arabidopsis*, RdDM has two main steps, i.e., biogenesis of 24-nt siRNAs and guidance of DRM2 to RdDM target by the siRNAs. Genomic regions showing RdDM-dependent DNA methylation always have siRNA enrichment[16]. To determine whether RDM15 is involved in siRNA biogenesis,

we compared *rdm15* mutants with Col−0, and identified 2808 RDM15-dependent siRNA clusters (hereafter referred to as RDM15 siRNAs) (Supplementary Data 2) that showed decreased 24-nt siRNA accumulation in *rdm15* mutants relative to Col−0. This result suggested that RDM15 is required for 24nt siRNA accumulation in vivo (Fig. 3a).

RdDM-dependent siRNAs can be classified into upstream siRNAs (siRNAs dependent on Pol IV only) and downstream siRNAs (siRNAs dependent on both Pol IV and Pol V)[16]. The upstream siRNAs are affected only in mutants defective in upstream RdDM components, such as *nrpd1*, whereas the downstream siRNAs are affected not only in these mutants but also in mutants defective in downstream RdDM components, such as *nrpe1* and *drm2* (Fig. 3b). To position RDM15 in the RdDM pathway, we examined the enrichment of RDM15-dependent siRNAs and found that RDM15 siRNA levels are lower in *nrpd1-4*, *nrpe1-12*, and

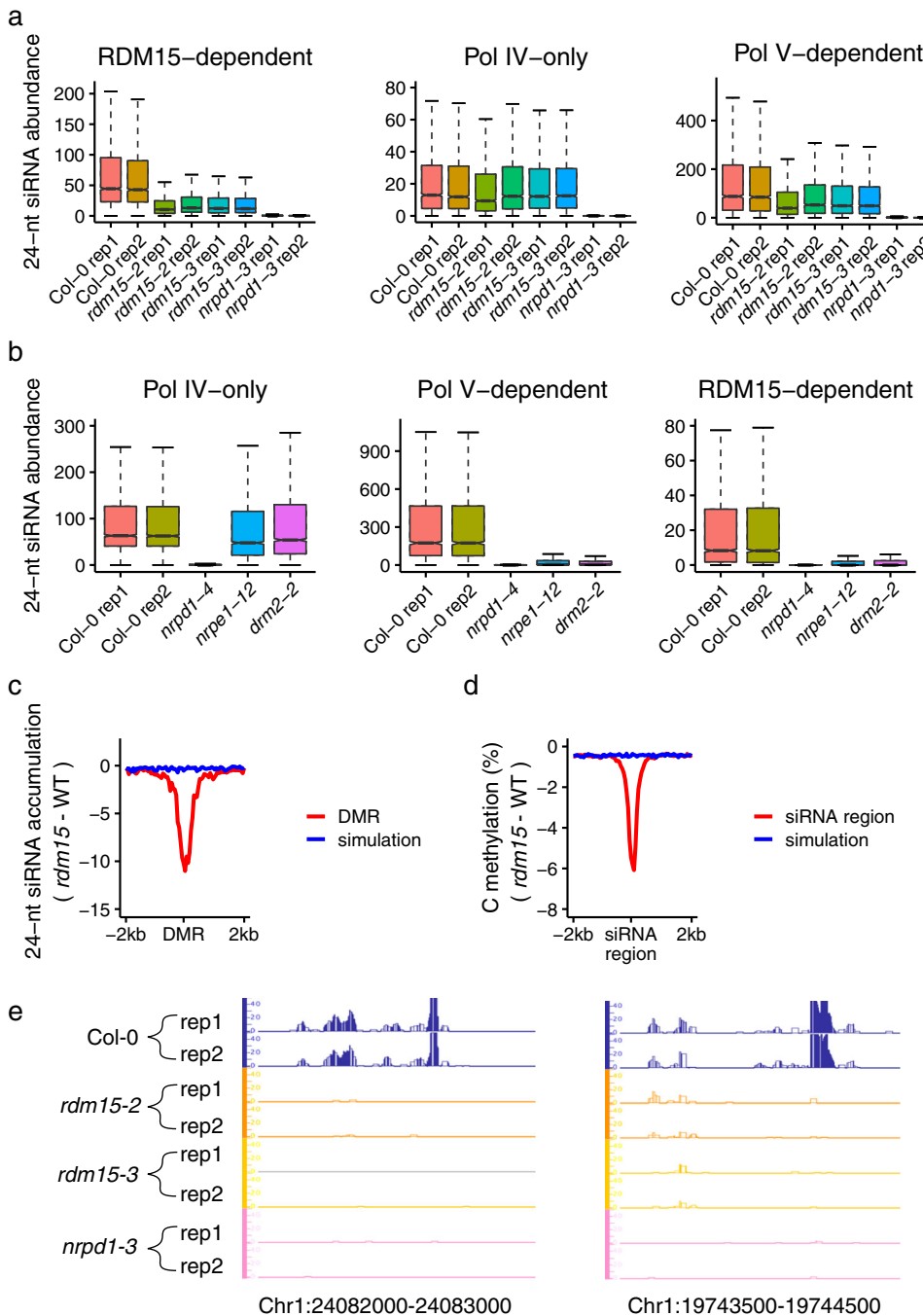

**Fig. 3 Dysfunction of *RDM15* abolishes RdDM-dependent 24-nt siRNA accumulation. a** Boxplots showing the abundance of 24-nt siRNAs in Col-0, *rdm15-2*, *rdm15-3*, and *nrpd1-3* with two replicates. The siRNA levels of RDM15-dependent siRNA cluster regions (left), Pol IV-only siRNA cluster regions (middle), and Pol V-dependent siRNA cluster regions (right) are shown. Compared to Col-0, RDM15 siRNAs were decreased in *rdm15* and *nrpd1-3* mutants, whereas Pol IV-only siRNAs (upstream RdDM siRNAs) were decreased in *nrpd1-3* but not in *rdm15*. **b** The abundance of 24-nt siRNAs is shown for Pol IV-only, Pol V-dependent, and RDM15-dependent siRNA regions in Col-0, *nrpd1-4*, *nrpe1-12*, and *drm2-2*. Pol IV-only and Pol V-dependent siRNAs represent upstream and downstream RdDM siRNAs, respectively. **c** Changes in 24-nt siRNA enrichment in *rdm15* relative to WT at *rdm15* hypo DMRs. **d** Changes in DNA methylation level, including mCG, mCHG, and mCHH, in *rdm15* relative to the WT at RDM15-dependent siRNA cluster regions. **e** IGB display of siRNA abundance of two RDM15-dependent siRNA regions in Col-0, *rdm15-2*, *rdm15-3*, and *nrpd1-3*. Two replicates are shown.

*drm2-2* compared to the WT (Fig. 3b). In addition, the Pol IV-only siRNAs are not significantly affected in *rdm15* mutants. In contrast, Pol V-dependent siRNAs are significantly reduced in *rdm15* mutants (Fig. 3a). These results showed that RDM15 is mainly required for the accumulation of downstream siRNAs in the RdDM pathway, suggesting that RDM15 functions in a downstream step of RdDM.

To further understand the relationship between the siRNAs and DNA methylation in *rdm15* mutants, we monitored changes in siRNA enrichment at hypo DMRs in *rdm15* mutants. The siRNA levels were clearly lower at hypo-DMRs in *rdm15* mutants than in the WT (Fig. 3c). Consistent with this result, the DNA methylation level, including mCG, mCHG, and mCHH, was decreased in regions of RDM15-dependent siRNA clusters (Fig. 3d

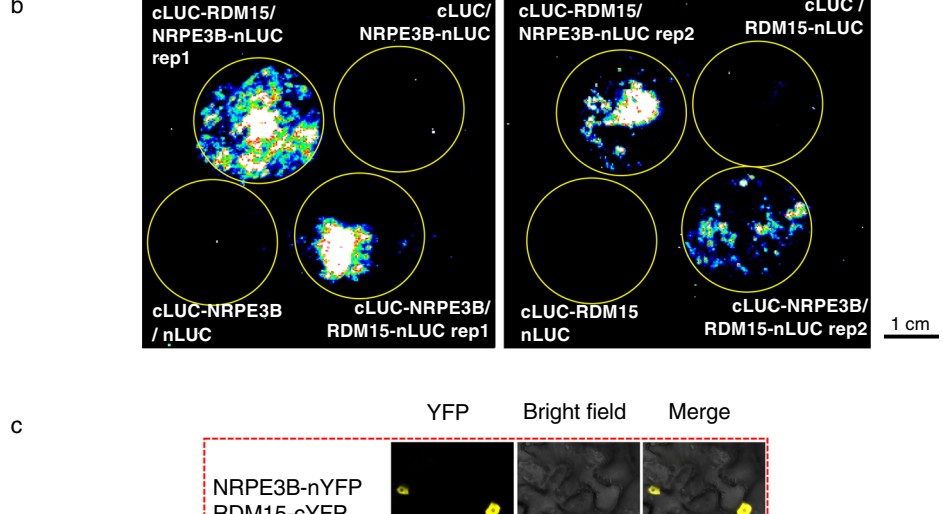

| RDM15 Purification (anti-FLAG) (*RDM15-3xFLAG/rdm15-3*) | | | | | | RDM15 Purification (anti-MYC) (*RDM15-3xMYC/rdm15-3*) | | | | | |
|---|---|---|---|---|---|---|---|---|---|---|---|
| Protein | Accession | Unique Peptides | | Coverage(%) | | Protein | Accession | Unique Peptides | | Coverage(%) | |
| | | Rep1 | Rep2 | Rep1 | Rep2 | | | Rep1 | Rep2 | Rep1 | Rep2 |
| RDM15 | AT4G31880.1 | 86 | 92 | 44.0 | 45.6 | RDM15 | AT4G31880.1 | 56 | 74 | 36.8 | 42.9 |
| Histone3 | AT5G65350.1 | 5 | 9 | 14.1 | 23.4 | Histone3 | AT5G65350.1 | 3 | 5 | 10.4 | 14.1 |
| NRPE3B | AT2G15400.1 | 3 | 7 | 5.2 | 10.1 | NRPE3B | AT2G15400.1 | 5 | 4 | 8.4 | 6.7 |

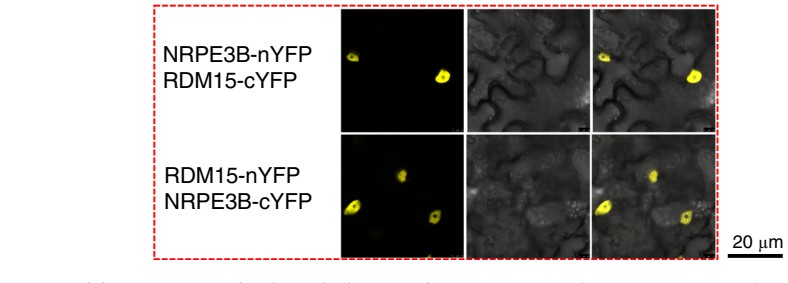

**Fig. 4 RDM15 physically interacts with NRPE3B, a subunit exclusive to Pol V. a** Detection of proteins associated with RDM15. LC-MS/MS was performed following immunoprecipitation of FLAG-tagged RDM15 and MYC-tagged RDM15. There were two biological replicates for each of the protein purifications. **b** Tests of RDM15-NRPE3B interaction by split luciferase complementation assay in tobacco leaves. Circles indicate leaf regions that were infiltrated with *Agrobacterium* strains containing the indicated constructs. (Scale bar = 1 cm) **c** Analyses of RDM15-NRPE3B protein interaction by bimolecular fluorescent complementation assay in tobacco leaves. Yellow fluorescence indicates positive protein interaction. (Scale bar = 20 μm).

and Supplementary Fig. 3b). The siRNA levels at several representative hypo-DMRs are shown in Fig. 3e and Supplementary Fig. 3c. These results suggested that the change of DNA methylation is associated with the change of siRNA levels at RDM15-dependent RdDM targets.

**RDM15 physically interacts with NRPE3B, a subunit exclusive to Pol V.** To further investigate how RDM15 affects RdDM, we identified RDM15-interacting proteins by performing immunoprecipitation (IP) followed by liquid chromatography–tandem mass spectrometry (LC–MS/MS) with *RDM15-3xMYC/rdm15-3* and *RDM15-3xFLAG/rdm15-3* plants (Supplementary Fig. 4a). WT plants that do not express *RDM15-3xMYC* or *RDM15-3xFLAG* were used as controls. We found that NRPE3B, a subunit exclusive to Pol V, was co-immunoprecipitated with anti-MYC antibodies and anti-FLAG antibodies in the *RDM15-3xMYC* and *RDM15-3xFLAG* transgenic plants, respectively, but not in the control plants (Fig. 4a and Supplementary Data 3). We validated the interaction between RDM15 and NRPE3B using a split luciferase complementation assay and a BiFC assay in tobacco leaves (Figs. 4b, c). In addition, we examined Pol V-dependent transcripts in *rdm15* mutants. The transcripts of *IGN25* and *IGN27*

were identified as Pol-V dependent transcripts in previous studies[27]. We found that the transcript levels of *IGN25* and *IGN27* were decreased in *rdm15-2* and *rdm15-3* mutants compared to WT (Supplementary Fig. 4b), suggesting that RDM15 is required for Pol V-dependent transcription. These data indicated that the RdDM function of RDM15 may involve its interaction with Pol V.

**The RDM15 Tudor domain specifically recognizes H3K4me1.** The results above show that RDM15 functions in the regulation of siRNA accumulation and DNA methylation at a subset of RdDM target loci. To further investigate how RDM15 functions, we analyzed its protein sequence and found two domains with known functions: an N-terminal ARM repeat, which is known to mediate protein–protein interactions, and a C-terminal Tudor domain, which functions as a histone mark reader (Fig. 5a). The Tudor domain has been found in many proteins related to epigenetic regulation and acts as a histone mark reader module that recognizes methylated lysine or arginine marks[28–30]. In addition, we found histone H3 in the affinity purification of RDM15 in vivo (Fig. 4a). In plants, histone mark readers sometimes have plant-specific binding targets and functions[13]. This prompted us to

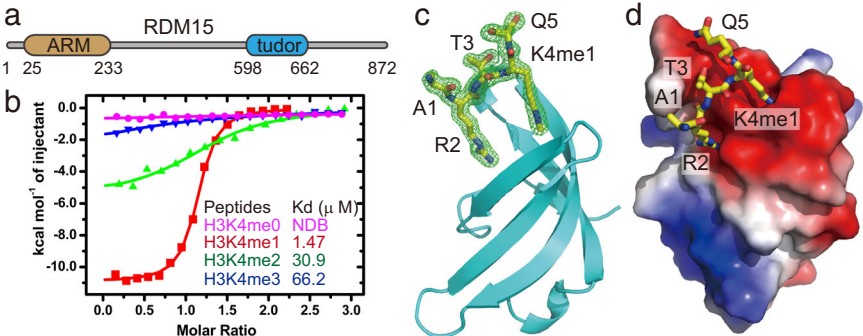

**Fig. 5 The RDM15 Tudor domain specifically recognizes H3K4me1 and the structure of its complex with the H3K4me1 peptide. a** Schematic representation of the domain architecture of RDM15. **b** ITC binding curves between the RDM15 Tudor domain and peptides with different methylation states on the H3K4 site; the curves indicate that RDM15 prefers the H3K4me1 mark over either lower (H3K4me0) or higher (H3K4me2 or H3K4me3) methylation states. NDB no detectable binding. **c** Overall structure of the RDM15 Tudor domain in complex with the H3K4me1 peptide, with RDM15 shown as cyan ribbon and the peptide shown as stick representation. The composite-omit electron density map at the 1σ level of the bound peptide is shown in a green mesh. **d** The RDM15 Tudor-H3K4me1 complex with RDM15 shown in an electrostatic surface and the peptide as stick representation. The interaction is mainly mediated by H3R2 and H3K4me1 residues, which insert their side chains in two negatively charged surface grooves of the Tudor domain.

explore the histone mark binding property of the RDM15 Tudor domain. In our assay using histone peptide arrays, which contained several hundred combinations of histone marks, the RDM15 Tudor domain showed significant binding to H3K4me1 and H4K20me1 marks (Supplementary Fig. 5a and Supplementary Data 4). We used isothermal titration calorimetry (ITC) to further confirm the binding and to measure the binding affinity between the RDM15 Tudor domain and different methyllysine-modified histone peptides. The H3K4me1 peptide yielded the strongest binding (1.47 μM) (Fig. 5b and Supplementary Table 1), while the H4K20me1 yielded a 15-fold lower binding affinity (22.9 μM) (Supplementary Fig. 5b and Supplementary Table 1), indicating that H4K20me1 is not the optimal binding partner for RDM15. In addition, researchers previously reported that the existence of H4K20 methylation in plants is controversial and that no in vivo H4K20 methylation could be detected by mass spectrometry[31]. We therefore focused on the interaction between the H3K4me1 mark and RDM15. We assessed the binding between the RDM15 Tudor domain and peptides with different H3K4 methylation states. ITC measurements clearly showed that the RDM15 Tudor domain bound much more strongly to the monomethylation state of H3K4 than to the unmethylated H3K4 (H3K4me0) or to the higher methylation states of H3K4 (H3K4me2 and H3K4me3) (Fig. 5b).

**The structure of RDM15 Tudor domain in complex with H3K4me1 peptide**. To further investigate the molecular mechanism underlying the specific recognition of the H3K4me1 mark by RDM15, we carried out structural studies. The crystal structure of the RDM15 Tudor domain in complex with the H3 (1-15) K4me1 peptide was determined using the SAD method and was refined to 1.7 Å resolution (Supplementary Table 2 and Fig. 5c). Although a 15-residue H3K4me1 peptide was used in the crystallization, only the H3A1 to H3Q5 segment of the peptide exhibited well-defined electron density and was built into the final model (Fig. 5c). The Tudor domain of RDM15 exhibits a canonical Tudor domain fold with five β-strands forming a twisted β-barrel structure that resembles the structure of other Tudor domains (Fig. 5c)[28].

Our results show that the RDM15 Tudor domain can specifically recognize H3K4me1. The main chain of the H3K4me1 peptide has a 'U'-shaped conformation such that the N-terminal H3A1 and the C-terminal H3Q5 extend out away from the peptide-protein binding interface (Fig. 5c). The side

chains of H3R2 and H3K4me1 form a pincer-like conformation, thereby anchoring on two adjacent negatively charged surface pockets of the Tudor domain and highlighting a significant structural and chemical complementarity (Fig. 5d). In detail, the H3R2 inserts its side chain into a negatively charged pocket and forms hydrogen-bonding interactions with Glu647 and Gln654 of RDM15, as well as a salt bridge interaction with Glu647 (Fig. 6a). For H3K4me1, the interactions can be divided into two parts. Three aromatic residues, Trp616, Tyr623, and Tyr641, form an aromatic cage to accommodate the methyl group of the monomethyllysine from one side, which is similar to the other canonical methyllysine readers[29,32] (Fig. 6b). On the other side, the two free protons of the monomethylammonium group are fully coordinated with Asp643 and Asp645 by hydrogen bonding and salt bridge interactions (Fig. 6b), representing a mono-methyllysine recognition mechanism (Supplementary Fig. 6 and see detailed analysis in the Discussion). The H3T3 positioned between H3R2 and H3K4me1 stretches its side chain against the protein and is not involved in the interaction with the RDM15 Tudor domain (Fig. 5c). In general, both H3R2 and H3K4me1 are specifically recognized by a hydrogen-bonding network and electrostatic interactions involving their charged side chains, as well as by an aromatic cage that captures the methylation modification (Fig. 6c). Mutations of residues of either the aromatic cage or of those involved in hydrogen bonding interactions with the peptide resulted in a significant decrease in the binding affinity (Fig. 6d, e), revealing that these residues are critical for the recognition of H3K4me1 by RDM15.

We expressed the mutated RDM15 (W616A/Y623A/D643A/D645A) incapable of binding to H3K4me1 in *rdm15-3* mutant plants, generating mutated *RDM15-3xMYC/rdm15-3* plants. We found that the H3K4me1 was co-immunoprecipitated by wild-type RDM15 but not by mutated RDM15 (Supplementary Fig. 5c), suggesting that the mutations in the Tudor domain interrupted the binding of H3K4me1 in vivo. We examined the methylation level using methylation-sensitive PCR at several RDM15-dependent RdDM loci and found that the wild type but not the mutated *RDM15* rescued the methylation phenotype in *rdm15-3* (Supplementary Fig. 5d), suggesting that when RDM15 is mutated to lose the binding of H3K4me1, the RdDM function of RDM15 is also impaired.

In the structure of the RDM15-H3K4me1 complex, the recognition of the H3 tail by the RDM15 Tudor domain highlights the specific recognition of H3R2 and H3K4me1, while other surrounding residues and the intervening H3T3 do not

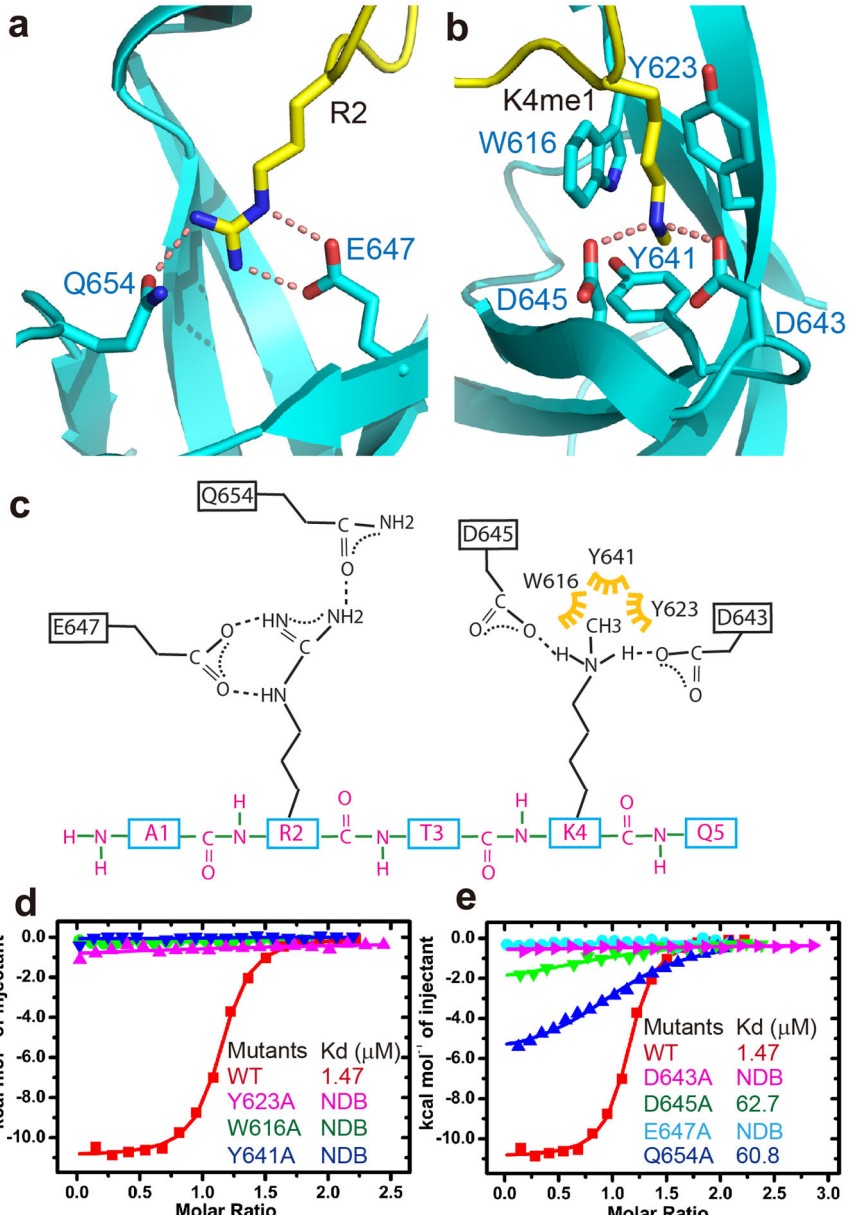

**Fig. 6 Details of the specific interaction between the RDM15 Tudor domain and the H3K4me1 peptide. a** The side chain of H3R2 is specifically recognized by Glu647 and Gln654 of the Tudor domain through a salt bridge and hydrogen-bonding interactions. The hydrogen bonds are highlighted with dashed red lines. **b** The methyl group of H3K4me1 is specifically accommodated within an aromatic cage formed by Trp616, Tyr623, and Tyr641. The two monomethylammonium protons of monomethyllysine form hydrogen bonding and salt bridge interactions with two negatively charged Asp643 and Asp645 residues. **c** A schematic representation of the intermolecular interactions between RDM15 and the H3K4me1 peptide. **d, e** The ITC binding curves between the H3K4me1 peptide and the RDM15 Tudor domain mutations showing that the disruption of the aromatic cage (**d**) or the residues involved in hydrogen bonding interactions (**e**) dramatically decreases the binding affinity.

contribute to the recognition. The RDM15 Tudor domain therefore recognizes an R̲X̲K̲me1 (X here stands for any residue) motif on histone tails. Among all four types of *Arabidopsis* histone proteins, there are two additional sites that contain the same motif and that might be recognized by the RDM15 Tudor domain: H2A R̲3T̲K̲5 and H4 R̲3G̲K̲5. A previous mass spectrometry study showed that the lysine residues of both of these two sites can be acetylated, but no methylation modification was identified in vivo[31]. It follows that although the RDM15 Tudor domain only recognizes two residues on the histone tail, the specific RXKme1 pattern only occurs in the H3K4 region, which ensures that recognition only occurs at H3K4me1, thereby explaining the sequence specificity.

**Characterization of chromatin targets of RDM15 binding**. We performed chromatin immunoprecipitation using native promoter-driven *RDM15-3xFLAG/rdm15-3* followed by high-throughput sequencing (ChIP-seq). We observed higher RDM15 protein enrichment in RDM15-dependent siRNA regions than in Pol IV-only siRNA regions (Fig. 7a), which is consistent with our above finding that RDM15 is required for the accumulation of RDM15 siRNAs but does not affect the accumulation of Pol IV-only siRNAs (Fig. 3). When we ranked RDM15 siRNA regions by the difference in siRNA accumulation between *rdm15* and the WT, we found that the change in siRNA accumulation (*rdm15* vs. the WT) was positively correlated with RDM15 protein enrichment; the change in siRNA accumulation in *nrpd1* vs. the WT

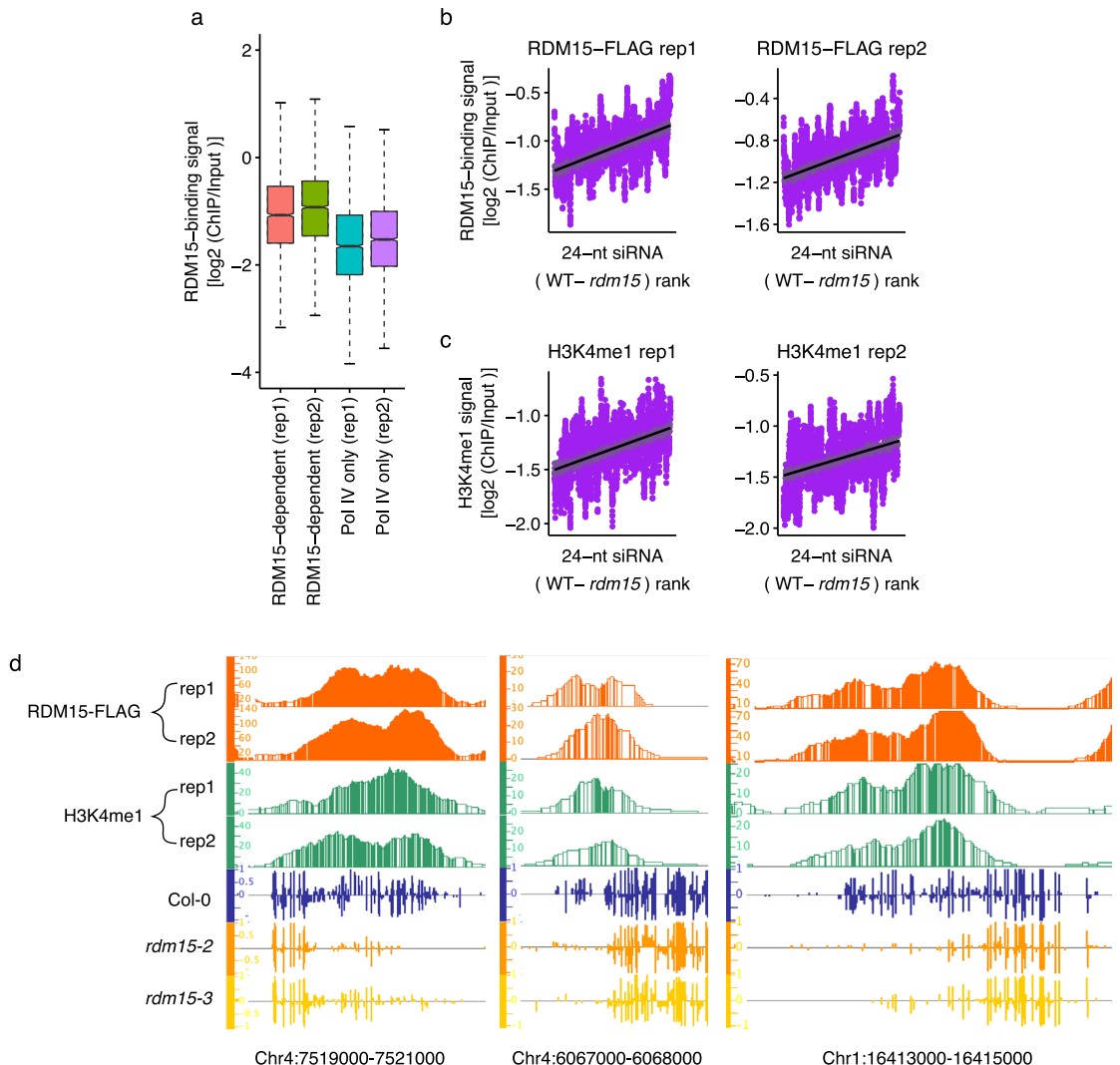

**Fig. 7 RDM15 binds to RDM15-dependent siRNA regions in the genome. a** Boxplots showing RDM15 protein enrichment in RDM15-dependent siRNA regions. Pol IV-only siRNA regions, which were not affected in *rdm15*, served as the control. In the boxplots, the center is the median of the data; upper bound of the box is an upper hinge, 75% quantile of the data; lower bound of the box is a lower hinge, 25% quantile of the data; minima is the smallest data point greater than or equal to lower hinge − 1.5 x interquartile range (IQR); maxima is the largest data point less than or equal to upper hinge + 1.5 x IQR. **b** Correlation of the change of siRNA accumulation (WT vs. *rdm15*) with RDM15 enrichment. RDM15-dependent siRNA regions were ranked by the change of siRNAs (WT vs. *rdm15*). **c** Correlation of the change of siRNA accumulation (WT vs. *rdm15*) with H3K4me1 enrichment. RDM15-dependent siRNA regions were ranked by the change of siRNAs (WT vs. *rdm15*). **d** IGB display of RDM15 and H3K4me1 protein enrichment and DNA methylation levels at several RDM15 binding targets.

served as the negative control (Fig. 7b and Supplementary Fig. 7a). These results suggest that the influence of RDM15 on siRNA accumulation is correlated with its protein enrichment. In addition, the distribution pattern of H3K4me1 but not of H3K4me2 and H3K4me3 was similar to that of RDM15 (Fig. 7c and Supplementary Fig. 7b), which is consistent with our finding of high-affinity binding of RDM15 to H3K4me1. The enrichment of RDM15 and H3K4me1 is shown for several representative RDM15 target regions in Fig. 7d and Supplementary Fig. 7c.

It has been reported that H3K4me1 is enriched in the gene body and flanking regions[33]. To investigate whether RDM15 can recognize H3K4me in genic regions, we examined the enrichment of RDM15 in genes. H3K4me1 and RDM15 showed similar distribution patterns around genes (Supplementary Fig. 7d), suggesting that RDM15 can also recognize H3K4me in genic regions. This result indicates that in addition to its function in RdDM, RDM15 may also have functions in regulating genes.

## Discussion

RdDM is an important DNA methylation pathway in plants and has been extensively studied during the past two decades. Several forward and reverse genetics screens, including the *ros1* suppressor screen in this study, were used to identify RdDM components[21,34–37]. Through this *ros1* suppressor screen, several RdDM mutants, such as *rdm1*, *rdm4*, and *ktf1*, were identified[22,23]. In the current study, an RdDM mutant, *rdm15*, was isolated and characterized. The methylome and siRNA analyses revealed that RDM15 is required for RdDM-dependent siRNA accumulation and DNA methylation at a subset of RdDM targets. In addition, the siRNA analysis and the interaction of RDM15 with Pol V suggested that RDM15 functions in the downstream steps of the RdDM pathway.

Several histone modifications, such as H3K9me2, have been reported to be associated with RdDM targets[12]. Here, we showed an association of H3K4me1 with RDM15-dependent RdDM

targets. The structural analysis demonstrated that RDM15 can specifically recognize and bind to H3K4me1 (Fig. 5). In addition, ChIP-seq results showed that the genomic distribution of RDM15 protein enrichment is similar to that of H3K4me1 but not H3K4me2/3 (Fig. 7 and Supplementary Fig. 7). H3K4me1 is usually thought to be associated with actively transcribed genes[38,39]. Among heterochromatic regions, Pol V-transcribed regions are presumably more active than untranscribed regions. We speculate that RDM15 may help recruit Pol V to a subset of RdDM targets by recognizing the H3K4me1 mark at these genomic regions. In addition, we found that RDM15 is also associated with H3K4me1 in genic regions, which are mostly not RdDM target regions. This result indicated that RDM15 may have an RdDM-independent function in these regions.

Many structural and functional studies have identified the general principles explaining the specific recognition of histone lysine methylation marks[29,30,32]. However, only a few studies have investigated the specific recognition of lower methyllysine, especially the monomethyllysine. L3MBTL1, a previously reported lower methylated lysine-specific binding protein, can specifically recognize mono- and dimethyllysine via the narrow and deep surface cavity of the protein (Supplementary Fig. 6a)[40]. In the recognition mode of L3MBTL1, almost the entire side chain of the methyllysine inserts into a deep cavity on the L3BMTL1 protein surface (Supplementary Fig. 6a). The size of the cavity therefore limits the size of the substrate. For L3MBLT1, the methyllysine must be in the lower methylation states of di- or monomethyllysine, representing a physical selection mechanism. The molecular mechanism for the specific recognition of H3K4me1 by the CW domain of *Arabidopsis* SDG8 was recently determined[41]. Like L3MBTL1, the SDG8 CW domain has a narrow cavity that accommodates the small side chain of H3K4me1 (Supplementary Fig. 6b), i.e., the recognition of H3K4me1 by SDG8 also involves physical selection[41]. In contrast, H3K4me1 is anchored within a surface groove of RDM15, with only the tip of the methyl group inserting into the canonical aromatic cage (Supplementary Fig. 6c). The specificity for the monomethylation of H3K4 is achieved by the full coordination of the two monomethylammonium protons by the two negatively charged residues, which increases the binding affinity for monomethylation. In other words, the specificity of H3K4me1 for RDM15 involves a chemical selection mechanism. The increase of the methylation state of the methyllysine from H3K4me1 to H3K4me2 or H3K4me3 would disrupt or even fully eliminate the hydrogen bonding interaction observed between RDM15 and the H3K4me1 side chain, resulting in a decreased binding affinity as revealed by our ITC data (Fig. 5b). This selection of a lower methylation state based on chemical features clearly differs from selection based on physical shape previously reported for L3MBTL1 and SDG8, and therefore represents a hydrogen bonding interaction-based but not a shaped-based mechanism for specific monomethyllysine recognition.

The Tudor domain can function in multiple modes, including single Tudor, double Tudor, tandem Tudor, or triple-linked Tudors like Spindlin1[28]. A single Tudor domain has been reported to be able to recognize H3K36me3, H4K20me2, and H3K4me3[14,28,42]. The Tudor domains of PHF1 and PHF19 have similar structures and recognition mechanisms and can specifically recognize the H3K36me3 mark; the structures of their complexes with H3K36me3 peptide have been reported[43–45], which enables us to perform a structure-based comparison. The RDM15 Tudor and PHF1 Tudor have a generally similar folding topology, with their aromatic cages occupying almost identical positions (Supplementary Fig. 6d)[45]. It is interesting that the H3K4me1 and H3K36me3 peptides occupy different positions, although their methylammonium groups occupy the same

position (Supplementary Fig. 6d), suggesting that the conserved aromatic cage of the Tudor domain serves as a docking site for methyllysine, with the diversified surrounding residues contributing to the histone mark specificity. PHF1 uses a four-residue aromatic cage to accommodate H3K36me3 (Supplementary Fig. 6d)[45]. Although Tyr47 and Phe65 of PHF1 adopt an almost identical conformation as Tyr623 and Tyr641 of RDM15, the Trp41 of PHF1 has a distinct torsion angle relative to Trp616 of RDM15, resulting in a broader pocket in PHF1, which is suitable for accommodating trimethyllysine; RDM15, in contrast, has a narrow pocket that only accommodates monomethyllysine. We also compared the structure of the RDM15-H3K4me1 complex with those of the JMJD2A double Tudor-H3K4me3 complex (PDB code: 2GFA) and SGF29 tandem Tudor-H3K4me3 complex (PDB code: 3MEA)[46,47]. Although it is difficult to superimpose the RDM15 Tudor to the corresponding Tudor domains of JMJD2A and SGF29 that bind to methylated histone peptides, the RDM15 Tudor can be well superimposed to the remaining Tudor domains of JMJD2A and SGF29 without peptide binding (Supplementary Fig. 6e), suggesting that the single Tudor and tandem Tudor provide different platforms to recognize histone marks with diversified modes and histone binding surface.

Our genetic and epigenomic analyses identified RDM15 as a factor required for siRNA accumulation and DNA methylation at a subset of genomic regions targeted by RdDM. Our biochemical and structural results demonstrated that RDM15 is a reader of H3K4me1, and revealed a hydrogen bonding interaction-based mechanism of recognition of lower methyllsine marks. Future work will determine whether RDM15 may affect the recruitment of Pol V to RdDM target regions and whether the mechanism of recognition of lower methyllysine mark by RDM15 may be applicable to other histone mark readers.

## Methods

**Plant materials and growth conditions**. The *Arabidopsis* C24 ecotype plant materials used in this study (WT, ros1-1, ros1-1 rdm15-1, ros1-1, nrpd1) carried a homozygous T-DNA insertion that contained the p35S-NPTII and pRD29A-LUC transgenes. T-DNA mutagenized ros1-1 populations were used for ros1-1 suppressor screening according to luciferase signals[22]. Plant materials of Columbia-0 (Col) ecotype included rdm15-2 (Salk_013481), rdm15-3 (Salk_024055), nrpd1-3 (SALK_128428C), and nrpe1-11 (SALK_029919C). Plants were grown in growth rooms at 22 °C with a 16–8 h light-dark cycle.

**Whole-genome bisulfite sequencing and data analyses**. Genomic DNA was extracted from 2-week-old seedlings using the Plant DNeasy mini kit (Qiagen) and was sent to the Core Facility for Genomics at the Shanghai Center for Plant Stress Biology (PSC) for whole-genome bisulfite sequencing (WGBS).

For WGBS data analysis, the raw data were trimmed using Trimmomatic[48] with parameters "LEADING:20 TRAILING:20 SLIDINGWINDOW:4:15 MINLEN:50". Only clean paired-end reads were mapped to the TAIR10 genome using tool BSMAP[49] with parameters "-m 0 -w 2" (other parameters were used with default values). To remove potential PCR duplicates, the "rmdup" command of SAMtools[50] was used. methratio.py script from BSMAP was used to extract methylation ratios from mapping results. The R package methylKit[26] was used to find differentially methylated regions (DMRs): the "tileMethylCounts" function was used to summarize methylated/unmethylated base counts over tilling windows (500 bp) across the genome. The "calculateDiffMeth" function was used to calculate differential methylation statistics between mutants and the WT, and the "getMethylDiff" function was used to select the DMRs with parameters "difference=8, qvalue=0.01". The selected 500-bp DMRs were merged using the "merge" command from bedtools[51]. For the PCA analysis (Supplementary Fig. 2b), the methylation levels were calculated in a 50-kb tiling window with a 10-kb step size across the genome. The two replicates of WGBS data for nrpd1-3 and nrpe1-11 were downloaded from the NCBI GEO database with accession numbers GSE44209[15] and GSE83802[52].

**Small RNA data analysis**. Total RNA was extracted from 2-week-old seedlings using the TRIzol (Invitrogen) method. RNA samples were separated on a PAGE gel, and the 18- to 30-nt fraction of the gel was cut for small RNA purification. Library preparation and sequencing were performed using Illumina reagents according to the manufacturer's instructions at the Genomics Core Facility of PSC. Small RNA data were analyzed according to ref. [53] with minor modifications. In

brief, the adapter was removed using the "fastx_clipper" command from FASTX-toolkit. After adapter sequences were trimmed, clean reads with sizes ranging from 18- to 31-nt were mapped to the *Arabidopsis* genome (TAIR10) using Bowtie[54] with parameters "-v 0 -k 10". Reads that overlapped with annotated tRNAs, rRNAs, snRNAs, or snoRNAs were excluded. Read counts were normalized to Reads Per Ten Million (RPTM) based on the total abundance of genome-matched small RNA reads. The "hits-normalized-abundance" (HNA) values were calculated by dividing the normalized abundance (in RPTM) for each small RNA hit, where a hit is defined as the number of loci at which a given sequence perfectly matches the genome[55]. The HNA values of small RNAs of all sizes within individual non-overlapping 200-bp windows throughout the whole genome were compared between the mutant and the WT. To focus on regions that show adequate small RNA accumulation in the WT plants, HNA values in the mutant and the WT samples were summed, and a cutoff of 25 was applied. Subsequently, only those 200-bp regions that showed ≥2-fold HNA reduction in the mutant were identified as small RNA-depleted regions. In Fig. 3a, small RNA data for Col−0, *rdm15-2*, and *rdm15-3* were sequenced for this project. Small RNA data for *nrpd1* were downloaded from the NCBI GEO database with accession number GSE83802[52]. In Fig. 3b, all of the small RNA data were downloaded from the NCBI GEO database with accession number GSE45368[16].

**Affinity purification and mass spectrometry**. The *RDM15* genomic sequence was fused in-frame to the *3xFLAG* tag and *3xMYC* tag, and the fused sequences were inserted into the pCambia1305 backbone and were transformed into *Arabidopsis* plants, respectively. A 5-g quantity of flower tissue was harvested from transgenic plants and ground in liquid nitrogen. The proteins were extracted in 25 ml of lysis buffer [50 mM Tris (pH 7.5), 150 mM NaCl, 5 mM MgCl₂, 10% glycerol, 0.1% Nonidet P-40, 0.5 mM DTT, 1 mM PMSF, and 250 µl of plant cell protease inhibitor (Sigma)]. After the proteins were extracted, 2 µl anti-Flag M2 (Sigma; F3165) or 5 µl anti-Myc (Sigma; M5546) was added and incubated at 4 °C for 2–3 h. The resins were then washed with the lysis buffer at least five times. The protein samples were sent to the Core Facility for Proteomics at the Shanghai Center for Plant Stress Biology (PSC) for affinity purification and mass spectrometry.

**Histone peptide array**. Histone peptide array assay was performed using the MODified Histone Peptide Array kit (Active Motif) according to the manufacturer's instructions. In brief, the slide was first blocked for 2 h at room temperature. After it was washed three times with TBST buffer [10 mM Tris·HCl (pH 7.5), 30 mM NaCl, and 0.05% Tween-20], the slide was incubated overnight with 10 mg of GST-fused purified RDM15 protein at 4 °C in 10 m of binding buffer [50 mM Hepes (pH 8.0), 150 mM NaCl, 2 mM DTT, and 0.05% Nonidet P-40]. After it was washed three additional times with TBST buffer, the slide was incubated with anti-GST primary antibody at 1/3000 dilution and secondary antibody at 1/5000 dilution in TBST buffer for 1 h at room temperature, respectively. After the washing, the slide was visualized using Lumi-light ECL substrate (Roche). The intensity of the blotted slide was analyzed according to the instructions of the analysis software of the MODified Histone Peptide Array.

**ChIP-seq analysis**. The ChIP assay was performed as previously described[56]. The 50 µl dynabeads Protein G (Invitrogen, cat# 10003D) were used for 2.5 µl antibody binding per tube and then we added the beads to the enriched nuclei faction for immunoprecipitation. The antibody for FLAG was anti-Flag M2 (Sigma; F3165).

The quality of the sequencing data was checked with FastQC. The paired-end reads were mapped to the TAIR10 genome of *Arabidopsis thaliana* (TAIR10) with bowtie2[54] with parameter "–very-sensitive–no-unal–no-mixed–no-discordant -k 5". To remove potential PCR duplicates, a markup from SAMtools[50] was used. After mapping, only uniquely mapped reads were retained for downstream analysis. The fragment number of interested regions was counted by featureCounts[57] with parameters "-p -O". The RDM15 enrichment (H3K4me1) signal was calculated as

$$\log_2[(1 + n\_ChIP)/N\_ChIP] - \log_2[(1 + n\_Input)/N\_Input],$$ where n_ChIP and n_Input represent the number of mapped ChIP and Input fragments in the interested regions, and N_ChIP and N_Input are the numbers of all mapped unique fragments. The public data for H3K4me2 and H3K4me3 were downloaded from the NCBI GEO database with accession number GSE113076.

**BiFC**. *Agrobacterium tumefaciens* strain GV3101 carrying constructs expressing *RDM15- cYFP*, *NRPE3B-nYFP* and *RDM15- nYFP*, or *NRPE3B-cYFP* were infiltrated into *N. benthamiana* leaves. After 2 d, the infiltrated areas were examined with a Leica TCS-SP8 microscope. This experiment was performed once.

**Split luciferase complementation assays**. Split luciferase complementation assays were performed in tobacco leaves. The coding sequences of RDM15 and NRPE3B were cloned into pCAMBIA-nLUC and pCAMBIA-cLUC vectors[58]. *A. tumefaciens* GV3101 carrying different constructs were infiltrated into *N. benthamiana* leaves. Two days after infiltration, luciferase activity was detected with a luminescence imaging system (Princeton Instruments). This experiment was performed once.

**Real-time RT-PCR analysis**. For real-time RT-PCR analysis, total RNA was extracted and contaminating DNA was removed with RNase-free DNase (RNeasy mini kit; Qiagen). mRNA (1 µg) was used for the first-strand cDNA synthesis with Takara RT-PCR Systems following the manufacturer's instructions. The cDNA reaction mixture was then diluted five times, and 1 µl was used as a template in a 15-µl PCR reaction with SYBR Green mix (Takara). Two or three independent biological replicates were used for analysis. Primer sequence information is listed in Supplementary Table 3.

**PCR-based DNA methylation analysis**. Genomic DNA was extracted using the Plant DNeasy mini kit (Qiagen) and 100 ng of the genomic DNA was digested with McrBC (NEB) in a 50 µl reaction system for 12 h. Then, a 1 µl digested DNA template was used for the target-specific PCR reaction. PCR products were analyzed by gel electrophoresis. Digestion without GTP was used as the control. Two independent biological replicates were used for analysis. Primer sequence information is listed in the Supplementary Table 3.

**Protein expression and purification**. The Tudor domain of *Arabidopsis thaliana* RDM15 (residues 598–662) was cloned into a pET-Sumo vector to fuse a hexahistidine and yeast Sumo tag to the target protein. The recombinant protein was expressed in *E. coli* strain BL21(DE3) by IPTG induction with a concentration of 0.3 mM. The protein was purified using a HisTrap (GE Healthcare) column. The His-Sumo tag was removed by ulp1 digestion followed by a second step with the HisTrap column. The target protein was further purified using a Superdex G75 column (GE Healthcare). The Se-Met labeled protein was expressed in Se-Met (Anatrace) containing M9 medium and was purified using the same protocol as used for the WT protein. All of the mutants were generated using a PCR-based mutagenesis method and were expressed and purified using the same protocol as used for the WT protein. All peptides used in this research were synthesized by GL Biochem (Shanghai).

**Crystallization, data collection, and structure determination**. Before crystallization, the purified RDM15 Tudor domain was concentrated to 20 mg/ml and mixed with H3(1-15) K4me1 peptide with a molar ratio of 1:4 at 4 °C for 30 min. The crystallization was conducted using the sitting-drop vapor diffusion method at 20 °C. The RDM15 Tudor domain in complex with H3K4me1 peptide was crystallized in a condition of 0.1 M Tris-HCl, pH 8.5, and 20% PEG 1000. The crystals were soaked in the reservoir solution supplemented with 15% glycerol and were flash cooled in liquid nitrogen for data collection. All the diffraction data were collected at the Shanghai Synchrotron Radiation Facility beamline BL17U1 and were processed using the HKL2000 program[59,60]. The structure of the RDM15 Tudor domain in complex with the H3K4me1 peptide was solved using the SAD method implemented in the Phenix program[61]. Model building was conducted using the Coot program[62]. The geometry of the model was analyzed using the MolProbity program[63]. All of the molecular graphics were generated using the program Pymol (Schrödinger, Inc.). A summary of diffraction data and structure refinement statistics is provided in Supplementary Table 2.

**Isothermal titration calorimetry**. ITC binding was measured using a Microcal PEAQ-ITC instrument (Malvern). The purified proteins were dialyzed against a buffer containing 100 mM NaCl, 20 mM HEPES, pH 7.0, and 2 mM β-mercaptoethanol and were diluted to a concentration of about 0.8–0.12 mM. The peptide was dissolved in the same buffer with a concentration of about 1.0–1.5 mM. All of the titrations were performed at 20 °C. The data were analyzed using the Origin 7.0 program.

**Reporting Summary**. Further information on research design is available in the Nature Research Reporting Summary linked to this article.

## Data availability
The data that support this study are available from the corresponding authors upon reasonable request. The sequencing data generated in the course of this study have been deposited in NCBI's Gene Expression Omnibus[64] and are accessible through GEO Series accession number GSE154302. The mass spectrometry data have been deposited in Integrated Proteome Resources with the accession number IPX0002945000. The structure of the *Arabidopsis* RDM15 Tudor domain in complex with an H3K4me1 peptide has been deposited into the RCSB Protein Data Bank with the accession code 7DE9. Source data are provided with this paper.

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

## Acknowledgements

We thank the staff at beamline BL17U1 at the Shanghai Synchrotron Radiation Facility for their help with data collection. This work was supported by the National Key R&D Program of China (2018YFD1000200 to Z.L., 2016YFA0503200 to J.D.), the Strategic Priority Research Program of the Chinese Academy of Sciences (XDB27040000 to Z.L., XDB27040101 to J.-K. Z.), Shenzhen Science and Technology Program (JCYJ20200109110403829 and KQTD20190929173906742 to J.D.) and Key Laboratory of Molecular Design for Plant Cell Factory of Guangdong Higher Education Institutes (2019KSYS006 to J.D.).

## Author contributions

Q.N., L.W., and Hu.Z. performed experiments. C.-G.D. performed histone peptide array. Z.G. and T.B. contributed to the mutation screen. Z.S. and L.C. contributed to the ITC and structure analysis. K.T. did the bioinformatics analysis. C.K. is involved in the discussion of the research. He.Z. performed the sequencing experiments. Q.N., Z.L., J.D., and J.-K.Z. designed the study, interpreted the data, and wrote the manuscript.

## Competing interests

The authors declare no competing interests.
