## [Peer Review File · Nature Communications]

REVIEWER COMMENTS

Reviewer #1 (Remarks to the Author):

In this paper, the authors identified RDM15 as a new factor of the RdDM pathway. Through extensive analysis, the authors identified RDM15 interacts with a Pol V subunit, and further verified its function in the RdDM pathway. In addition, they found that RDM15 contains a Tudor domain that specifically recognizes H3K4me1 methylation. The structure feature of the lower methylation state (H3K4me1) recognition by RDM15 Tudor is very novel. In addition, the connection between H3K4me1 and RdDM is also a very interesting finding. This paper is a very solid original research, the authors explore the functions of their newly identified RdDM factor RDM15 through different angles, so the RdDM-related functions of RDM15 are convincing. In addition, they found the connection between histone modification H3K4me1 and RdDM, which is very novel and would complement the connection between histone modification and DNA methylation in plants. So I think this paper meets the stand of Nature Communications and would be of broad interests.

Several points are listed below to further improve the paper.

1. In the introduction part, all the full names of the proteins introduced are written in Capital letters. Is that required, or is there a standard way of writing?
2. In figure 1b, the scales in the y axis are misleading. Because 1-10, 10-100 and 100-1000 are of the same lengths. There should be a better way to represent the data.
3. A more detailed list of the parameters measured by ITC should be shown either in the method section, or in a supplementary table.
4. This paper contains two major parts. The first part identified RDM15 as a new RdDM factor, which functions through interaction with Pol V. Data for this part are very solid. The second part identified RDM15 as a special H3K4me1 reader, which potentially connects H3K4me1 modification to RdDM pathway. For this part, the connection between H3K4me1 modification and RDM15 function in RdDM are not strong enough. The authors may need to do some mutational analysis based on structural information to further clarify the connection. For example, when RDM15 are mutated to lose the binding of H3K4me1, will the RdDM function of RDM15 affected? Will the recruitment of RDM15 affected? The functional relevance of H3K4me1 in plants are not well known, information from such analysis would further clarify whether H3K4me1 modification is related to RdDM function of RDM15 or not, which would further strengthen the findings in the second part.

Reviewer #2 (Remarks to the Author):

In this study, Niu and colleagues characterize RDM15, a new component of the RNA-directed process of DNA methylation in plants. The authors found that RDM15 binds to NRPE3B and additionally to histone H3K4me1 through its Tudor domain. The authors determined the crystal structure of the RDM15 Tudor domain in complex with H3K4me1 peptide, which clearly explains selectivity of this protein toward monomethylated lysine and the RxKme1 motif. Overall, the manuscript describes novel and significant

for the plant chromatin biology findings, contains excellent quality data, and has justifiable conclusions.

A few minor comments:

Are the aromatic cage mutants still folded?

It might be informative to also compare the RDM15 Tudor domain to the tandem Tudor domains that recognize methylated H3K4.

Please add the citation Musselman, 2012, NSMB when discuss overall histone/epigenetic readers.

Reviewer #3 (Remarks to the Author):

This study identified a new regulator of RNA-directed DNA methylation by a forward genetic screen. Based on small RNA deep sequencing and bisulfite sequencing, the study suggests that RDM15 is a canonical regulator involved in RNA-directed DNA methylation and acts as a downstream step of RdDM. The notion was further supported by the interaction of RDM15 with NRPE3B, a subunit of Pol V. By biochemical and structural analyses, the study showed that RDM15 can bind to the histone modification H3K4me1. The results demonstrated that RDM15 is an RdDM component and revealed a possible role of RDM15 in RdDM. To date, most of RdDM regulators were identified by forward genetic screen and it is difficult to identify a new regulator. The finding will help understand how DNA methylation is established at specific chromatin loci by RdDM. I feel that some analyses need to be improved. It is necessary to clarify whether RDM15 contributes to DNA methylation through facilitating siRNA production or not. More comments contain:

1. The statement related to Figure 3C, 3D is misleading and needs to be revised. The result shown above indicated that RDM15 is only required for the accumulation of downstream siRNAs but not upstream Pol IV-only dependent siRNAs. This suggests that involvement of RDM15 in siRNA accumulation is more likely to be caused by its effect on DNA methylation and is not required for its function in DNA methylation. For those Pol IV-only dependent siRNA loci, involvement of Pol V is dispensable for siRNA accumulation but is necessary for DNA methylation. Similarly, as a downstream component of RdDM, RDM15 is most likely to be also involved in DNA methylation at those pol IV-only dependent siRNA loci. Figure 3C and 3D showed overall DNA methylation or siRNA levels at RDM15-dependent RdDM targets. This analysis can not differentiate the Pol IV-only dependent siRNA loci and Pol V-dependent siRNA loci. Therefore, the last sentence of this section "These results suggest that RDM15 influences DNA methylation at RdDM target regions by regulating siRNA accumulation" is not accurate and needs to be revised.

2. It is interesting to find that RDM15 interacts with NRPE3B, a subunit of Pol V. This is consistent with the finding that RDM15 acts a downstream step of RdDM. It is possible that RDM15 may be necessary for association of Pol V with chromatin.

3. Because RDM15 interacts with the Pol V subunit NRPE3B, it is reasonable to predict that RDM15 is required for Pol V-dependent transcription. It is necessary to detect the Pol V-dependent transcript level in the *rdm15* mutant.

4. The specific interaction of RDM15 with H3K4me1 was found by ITC. Because the crystal structure of RDM15-H3K4me1 was produced, it is necessary to discuss why RDM15 specifically interacts with H3K4me1 but not with H3K4me2 and H3K4me3.

5. In line 1, page 6, the first line of the "RDM15 is required for RdDM-dependent DNA methylation" section, the effect of RDM15 needs to be changed to the effect of *rdm15*.

6. Figure 2B and 2C were not correctly labeled and were also not correctly cited in the text.

7. In Figure 3a, the result showed the Pol IV only dependent siRNA levels in *rdm15* mutants but did not show how Pol V-dependent siRNA levels were affected in *rdm15* mutants. The result need to be showed to determine whether RDM15 is necessary for Pol V-dependent siRNA accumulation. Otherwise, it is not known how much RDM15 contributes to Pol V-dependent siRNA accumulation.

8. The histone H3 was also identified by affinity purification of RDM15 but was not mentioned (Figure 4A). Because RDM15 was shown to interact with H3K4me1 as determined by in vitro assay, I wonder is the histone H3 is enriched with H3K4me1. If it is true, it will support the conclusion of the study.

9. In Figure 7a, the enrichment of RDM15 was compared between RDM15-dependent siRNA loci and Pol IV-only dependent siRNA loci. Because the downstream RdDM components such as Pol V can affect DNA methylation even when they did not affect siRNA accumulation. Because RDM15 interacts with Pol V subunit NRPE3B, RDM15 may take effect in a similar manner with Pol V. I suggest that the enrichment of RDM15 is compared between RDM15-dependent DMRs and RDM15-independent RdDM DMRs.

10. The RDM15 ChIP-seq data should be analyzed in more details. For example, how many RDM15 peaks were identified by the ChIP-seq analysis. how RDM15 ChIP-seq signals are distributed in genes, TEs, and intergenic regions. Considering that H3K4me1 is enriched in gene body but not in TEs and DNA methylated regions, it is necessary to clarify how RDM15 specially recognizes RdDM loci with H3K4me1 but does not recognized other chromatin loci with H3K4me1.

11. It is interesting to find that RDM15 binds to H3K4me1. It is necessary to determine whether the binding is required for the function of RDM15 in RdDM in vivo.

12. The raw data GSE154302 shown in the manuscript are not available online in GEO.

We thank the reviewers for their comments and suggestions. Our responses to the comments are highlighted in red in the following.

REVIEWER COMMENTS

Reviewer #1 (Remarks to the Author):

In this paper, the authors identified RDM15 as a new factor of the RdDM pathway. Through extensive analysis, the authors identified RDM15 interacts with a Pol V subunit, and further verified its function in the RdDM pathway. In addition, they found that RDM15 contains a Tudor domain that specifically recognizes H3K4me1 methylation. The structure feature of the lower methylation state (H3K4me1) recognition by RDM15 Tudor is very novel. In addition, the connection between H3K4me1 and RdDM is also a very interesting finding. This paper is a very solid original research, the authors explore the functions of their newly identified RdDM factor RDM15 through different angles, so the RdDM-related functions of RDM15 are convincing. In addition, they found the connection between histone modification H3K4me1 and RdDM, which is very novel and would complement the connection between histone modification and DNA methylation in plants. So I think this paper meets the stand of Nature Communications and would be of broad interests.

Several points are listed below to further improve the paper.

We thank the reviewer for the positive comments.

1. In the introduction part, all the full names of the proteins introduced are written in Capital letters. Is that required, or is there a standard way of writing?

Answer: The full names of plant proteins are commonly written in capital letters.

2. In figure 1b, the scales in the y axis are misleading. Because 1-10, 10-100 and 100-1000 are of the same lengths. There should be a better way to represent the data.

Answer: We thank the reviewer for pointing out the problem, and have changed the scale to log10 in the revised manuscript.

3. A more detailed list of the parameters measured by ITC should be shown either in the method section, or in a supplementary table.

Answer: We have listed more parameters measured by ITC in the Supplementary Table 5.

4. This paper contains two major parts. The first part identified RDM15 as a new RdDM factor, which functions through interaction with Pol V. Data for this part are very solid. The second part identified RDM15 as a special H3K4me1 reader, which potentially connects H3K4me1 modification to RdDM pathway. For this part, the connection between H3K4me1

modification and RDM15 function in RdDM are not strong enough. The authors may need to do some mutational analysis based on structural information to further clarify the connection. For example, when RDM15 are mutated to lose the binding of H3K4me1, will the RdDM function of RDM15 affected? Will the recruitment of RDM15 affected? The functional relevance of H3K4me1 in plants are not well known, information from such analysis would further clarify whether H3K4me1 modification is related to RdDM function of RDM15 or not, which would further strengthen the findings in the second part.

Answer: According to the reviewer's suggestion, we expressed mutated RDM15 in *rdm15-3* (*mRDM15-3xMYC/rdm15-3*). The mutated RDM15 carries the mutations of W616A/Y623A/D643A/D645A, which abolish the binding to H3K4me1 according to ITC results in figure 6. We examined the methylation level at several RDM15-dependent RdDM loci, and found that the wild type RDM15 can rescue the methylation phenotype in *rdm15-3*, but the mutated RDM15 cannot rescue the methylation phenotype, even though the mutated protein is expressed (Supplementary Fig. 5c), suggesting that when RDM15 was mutated to lose the binding of H3K4me1, the RdDM function of RDM15 was impaired.

Reviewer #2 (Remarks to the Author):

In this study, Niu and colleagues characterize RDM15, a new component of the RNA-directed process of DNA methylation in plants. The authors found that RDM15 binds to NRPE3B and additionally to histone H3K4me1 through its Tudor domain. The authors determined the crystal structure of the RDM15 Tudor domain in complex with H3K4me1 peptide, which clearly explains selectivity of this protein toward monomethylated lysine and the RxKme1 motif. Overall, the manuscript describes novel and significant for the plant chromatin biology findings, contains excellent quality data, and has justifiable conclusions.

We thank the reviewer for the positive comments.

A few minor comments:

Are the aromatic cage mutants still folded? It might be informative to also compare the RDM15 Tudor domain to the tandem Tudor domains that recognize methylated H3K4. Please add the citation Musselman, 2012, NSMB when discuss overall histone/epigenetic readers.

Answer: Usually, the unfolded protein may heavily aggregate or precipitate. All the RDM15 aromatic cage mutants behave well like the wild type protein with high solubility and monomeric form as shown in our purification step. So, we believe that these mutants are still folded.

We compared our RDM15-H3K4me1 and the JMJD2A double tudor-H3K4me3 complex (PDB code: 2GFA) and SGF29 tandem tudor-H3K4me3 complex (PDB code: 3MEA), which showed that they use different mode and surface to interact with histone peptide. This has been added in the Supplementary Fig. 6e.

We apologize for missing this milestone review and have added it in the manuscript.

Reviewer #3 (Remarks to the Author):

This study identified a new regulator of RNA-directed DNA methylation by a forward genetic screen. Based on small RNA deep sequencing and bisulfite sequencing, the study suggests that RDM15 is a canonical regulator involved in RNA-directed DNA methylation and acts as a downstream step of RdDM. The notion was further supported by the interaction of RDM15 with NRPE3B, a subunit of Pol V. By biochemical and structural analyses, the study showed that RDM15 can bind to the histone modification H3K4me1. The results demonstrated that RDM15 is an RdDM component and revealed a possible role of RDM15 in RdDM. To date, most of RdDM regulators were identified by forward genetic screen and it is difficult to identify a new regulator. The finding will help understand how DNA methylation is established at specific chromatin loci by RdDM. I feel that some analyses need to be improved. It is necessary to clarify whether RDM15 contributes to DNA methylation through facilitating siRNA production or not. More comments contain:

We thank the reviewer for the positive comments.

1. The statement related to Figure 3C, 3D is misleading and needs to be revised. The result shown above indicated that RDM15 is only required for the accumulation of downstream siRNAs but not upstream Pol IV-only dependent siRNAs. This suggests that involvement of RDM15 in siRNA accumulation is more likely to be caused by its effect on DNA methylation and is not required for its function in DNA methylation. For those Pol IV-only dependent siRNA loci, involvement of Pol V is dispensable for siRNA accumulation but is necessary for DNA methylation. Similarly, as a downstream component of RdDM, RDM15 is most likely to be also involved in DNA methylation at those pol IV-only dependent siRNA loci. Figure 3C and 3D showed overall DNA methylation or siRNA levels at RDM15-dependent RdDM targets. This analysis can not differentiate the Pol IV-only dependent siRNA loci and Pol V-dependent siRNA loci. Therefore, the last sentence of this section "These results suggest that RDM15 influences DNA methylation at RdDM target regions by regulating siRNA accumulation" is not accurate and needs to be revised.

Answer: According to the reviewer's suggestion, we rephrased the last sentence in the revised manuscript as following "These results suggested that the change of DNA methylation is associated with the change of siRNA level at RDM15-dependent RdDM targets"

2. It is interesting to find that RDM15 interacts with NRPE3B, a subunit of Pol V. This is consistent with the finding that RDM15 acts a downstream step of RdDM. It is possible that RDM15 may be necessary for association of Pol V with chromatin.

Answer: We agree with the reviewer that RDM15 may be necessary for association of Pol V with chromatin. An examination of the effect of RDM15 on the association of Pol V with chromatin would require Pol V-tag/*rdm15* plant materials, which we could not generate in a short time. However, in the revised manuscript, we examined the Pol V-

dependent transcript levels in *rdm15*, and found that the transcript levels of *IGN25* and *IGN27*, which were identified as Pol V transcripts in previous studies, were decreased in *rdm15* mutants compared to Col-0 wild type (Supplementary Fig. 4b). This result supports that RDM15 is necessary for Pol V function.

3. Because RDM15 interacts with the Pol V subunit NRPE3B, it is reasonable to predict that RDM15 is required for Pol V-dependent transcription. It is necessary to detect the Pol V-dependent transcript level in the *rdm15* mutant.

Answer: As suggested by the reviewer, we examined Pol V-transcript levels in *rdm15-2* and *rdm15-3* mutants. As shown in Supplementary Fig. 4b, the transcripts of *IGN25* and *IGN27*, which were identified as Pol V transcripts in previous studies, were downregulated in *rdm15* mutants compared to Col-0 wild type.

4. The specific interaction of RDM15 with H3K4me1 was found by ITC. Because the crystal structure of RDM15-H3K4me1 was produced, it is necessary to discuss why RDM15 specifically interacts with H3K4me1 but not with H3K4me2 and H3K4me3.

Answer: We have included a short discussion about this: "The increase of the methylation state of the methyllysine from H3K4me1 to H3K4me2 or H3K4me3 would disrupt or even fully eliminate the hydrogen bonding interaction observed between RDM15 and the H3K4me1 side chain, resulting in a decreased binding affinity as revealed by our ITC data (Fig. 5b)."

5. In line 1, page 6, the first line of the "RDM15 is required for RdDM-dependent DNA methylation" section, the effect of RDM15 needs to be changed to the effect of *rdm15*.

Answer: In the revised manuscript, we changed "To characterize the effect of RDM15 on DNA methylation..." to "To characterize the effect of *rdm15* on DNA methylation...".

6. Figure 2B and 2C were not correctly labeled and were also not correctly cited in the text.

Answer: In figure 2b, we added "*rdm15* hypo DMRs" above the figure to indicate that the DNA methylation level shown is for *rdm15* hypo DMRs. In figure 3C, we changed "*rdm15* DMRs, *nrpd1-3* DMRs, and *nrpe1-11* DMRs" to "*rdm15* hypo DMRs, *nrpd1-3* hypo DMRs, and *nrpe1-11* hypo DMRs". In the revised manuscript, we cited figure 2b and 2c in the following part: "Similar to known RdDM targets, *rdm15* DMRs show DNA hypomethylation in all three sequence contexts (mCG, mCHG, and mCHH) compared to wild type (Fig. 2b), although the methylation level in *rdm15* mutants is not as low as in *nrpd1-3* (Fig. 2b). We further analyzed the overlap between RDM15 targets and known RdDM targets. As shown in Fig. 2c, 94% (1,269/1,354) and 95% (1,281/1,354) of *rdm15* hypo DMRs overlap with the *nrpd1* and *nrpe1* hypo DMRs, respectively."

7. In Figure 3a, the result showed the Pol IV only dependent siRNA levels in *rdm15* mutants but did not show how Pol V-dependent siRNA levels were affected in *rdm15* mutants. The result need to be showed to determine whether RDM15 is necessary for

Pol V-dependent siRNA accumulation. Otherwise, it is not known how much RDM15 contributes to Pol V-dependent siRNA accumulation.

Answer: Following the reviewer's suggestion, we examined Pol V-dependent siRNA accumulation in *rdm15* mutants and found that Pol V-dependent siRNA levels were reduced in *rdm15* mutants (Fig. 3a). We added this result in the revised manuscript: "In addition, the Pol IV-only siRNAs are not significantly affected in *rdm15* mutants. In contrast, Pol V-dependent siRNAs are significantly reduced in *rdm15* mutants (Fig. 3a)".

8. The histone H3 was also identified by affinity purification of RDM15 but was not mentioned (Figure 4A). Because RDM15 was shown to interact with H3K4me1 as determined by in vitro assay, I wonder is the histone H3 is enriched with H3K4me1. If it is true, it will support the conclusion of the study.

Answer: Following the reviewer's suggestion, in the revised manuscript, we mentioned that "In addition, we found that histone H3 was identified in the affinity purification of RDM15 *in vivo* (Fig. 4a)". Although we didn't show whether the histone H3 is enriched with H3K4me1, the ITC, structural analysis, and CHIP-seq analysis all supported the binding of RDM15 with H3K4me1.

9. In Figure 7a, the enrichment of RDM15 was compared between RDM15-dependent siRNA loci and Pol IV-only dependent siRNA loci. Because the downstream RdDM components such as Pol V can affect DNA methylation even when they did not affect siRNA accumulation. Because RDM15 interacts with Pol V subunit NRPE3B, RDM15 may take effect in a similar manner with Pol V. I suggest that the enrichment of RDM15 is compared between RDM15-dependent DMRs and RDM15-independent RdDM DMRs.

Answer: Following the reviewer's suggestion, we compared the enrichment of RDM15 between RDM15-dependent targets and RDM15-independent RdDM targets. As shown in the following figure, RDM15 enrichment is higher in RDM15-dependent targets compared to RDM15-independent RdDM targets, which is consistent with our result from the comparison between RDM15-dependent loci and Pol IV-only loci. In the manuscript, to be consistent with our siRNA analysis in figure 3, the comparison between RDM15-dependent loci and Pol IV-only loci is kept: "We observed higher RDM15 protein enrichment in RDM15-dependent siRNA regions than in Pol IV-only siRNA regions (Fig. 7a), which is consistent with our above finding that RDM15 is required for the accumulation of RDM15 siRNAs but does not affect the accumulation of Pol IV-only siRNAs (Fig. 3)."

10. The RDM15 ChIP-seq data should be analyzed in more details. For example, how many RDM15 peaks were identified by the ChIP-seq analysis. how RDM15 ChIP-seq signals are distributed in genes, TEs, and intergenic regions. Considering that H3K4me1 is enriched in gene body but not in TEs and DNA methylated regions, it is necessary to clarify how RDM15 specially recognizes RdDM loci with H3K4me but does not recognized other chromatin loci with H3K4me1.

Answer: We agree with the reviewer that most of the H3K4me1 mark is enriched in gene body but not in TEs and DNA methylated regions. In the revised manuscript, we examined the RDM15 ChIP-seq signal in genes, and found that both H3K4me1 and RDM15 were preferentially distributed in the 3' end of genes (Supplementary Fig. 7c), suggesting that RDM15 also binds to the H3K4me1 mark in genic regions. This result indicates that in addition to its role in RdDM, RDM15 might also have functions in regulating genes, which requires further studies in the future.

11. It is interesting to find that RDM15 binds to H3K4me1. It is necessary to determine whether the binding is required for the function of RDM15 in RdDM in vivo.

Answer: According to the reviewer's suggestion, we expressed mutated RDM15 in *rdm15-3*, generating *mRDM15-3xMYC/rdm15-3* plants. The mutated RDM15 carries the mutations of W616A/Y623A/D643A/D645A, which can abolish the binding of H3K4me1 according to ITC results in figure 6. We examined the methylation level at several RDM15-dependent RdDM loci, and found that the wild type RDM15 can rescue the methylation phenotype in *rdm15-3*, but the mutated RDM15 cannot rescue the methylation phenotype (Supplementary Fig. 5C), suggesting that when RDM15 was mutated to lose the binding of H3K4me1, the RdDM function of RDM15 was also impaired.

12. The raw data GSE154302 shown in the manuscript are not available online in GEO. The raw data GSE154302 shown in the manuscript are not available online in GEO.

Answer: The data can be viewed at <https://www.ncbi.nlm.nih.gov/geo/query/acc.cgi?acc=GSE154302> with secure token: elupcyimpfihrib
We will make it public after the manuscript is accepted.

REVIEWER COMMENTS

Reviewer #1 (Remarks to the Author):

The authors have fully addressed my concerns. I think this paper meets the standard of a Nature Communications paper.

Reviewer #3 (Remarks to the Author):

The study suggests that the binding of the RDM15 Tudor domain to H3K4me1 is required for RNA-directed DNA methylation. However, whether the RDM15 Tudor domain is required for the binding of RDM15 to H3K4me1 was not demonstrated *in vivo*. Therefore, I suggest to test the hypothesis in my previous comments. In the revised manuscript, the RDM15 transgene with mutation in Tudor domain was transformed into the *rdm16* mutant for complementation assay and indicated that the DNA methylation defect in the *rdm15* mutant was not complemented by the mutated transgene. Of course, this experiment supports that the tudor domain is required for DNA methylation. However, the binding the Tudor domain to H3K4me1 was only tested in *in vitro* assays. The effect of the Tudor mutation on the binding of RDM15 to H3K4me1 needs to be tested *in vivo*. Otherwise, the current results are not sufficient for supporting the conclusion. Because the transgenic plants carrying the RDM15 transgene with the Tudor mutation were already available as shown in the current study, I suggest to perform ChIP-seq or ChIP-PCR in RDM15 transgenic plants with and without the Tudor mutation to test the effect of the tudor mutation on the binding to RDM15 to chromatin.

Moreover, the H3K4me1 mark has been previously detected by ChIP-seq at whole-genome level in a previous study (Inagaki et al., EMBO J., 2017). The study showed that H3K4me1 is mainly located in genic regions. Because most of genic regions are not methylated and are unlikely to be RdDM loci, it is necessary to determine the relationship between RDM15 ChIP-seq signals and H3K4me1 signals at the whole-genome level. The H3K4me1 ChIP-seq was reanalyzed in this study in order to compare the H3K4me1 signals and RdDM loci. However, it is necessary to analyze to what extent the RDM15 peaks overlap with H3K4me1 peaks at the whole-genome level and to determine whether RDM15 peaks overlap with H3K4me1 peaks at the genic regions that are not targeted by RdDM. These analyses are important for understanding whether RDM15 is specifically involved in RdDM or extensively associated with H3K4me1 at the whole-genome level. If RDM15 is enriched at all H3K4me1-enriched genic regions at the whole-genome level, RDM15 is unlikely to be a canonical component that is specifically required for RdDM.

We thank the reviewer for their comments. We will response the comments in the following with our response highlighted in red.

REVIEWER COMMENTS

Reviewer #1 (Remarks to the Author):

The authors have fully addressed my concerns.I think this paper meets the standard of a Nature Communications paper.

Answer: We thank the reviewer for the positive comments.

Reviewer #3 (Remarks to the Author):

The study suggests that the binding of the RDM15 Tudor domain to H3K4me1 is required for RNA-directed DNA methylation. However, whether the RDM15 Tudor domain is required for the binding of RDM15 to H3K4me1 was not demonstrated in vivo. Therefore, I suggest to test the hypothesis in my previous comments. In the revised manuscript, the RDM15 transgene with mutation in Tudor domain was transformed into the *rdm16* mutant for complementation assay and indicated that the DNA methylation defect in the *rdm15* mutant was not complemented by the mutated transgene. Of course, this experiment supports that the tudor domain is required for DNA methylation. However, the binding the Tudor domain to H3K4me1 was only tested in in vitro assays. The effect of the Tudor mutation on the binding of RDM15 to H3K4me1 needs to be tested in vivo. Otherwise, the current results are not sufficient for supporting the conclusion. Because the transgenic plants carrying the RDM15 transgene with the Tudor mutation were already available as shown in the current study, I suggest to perform ChIP-seq or ChIP-PCR in RDM15 transgenic plants with and without the Tudor mutation to test the effect of the tudor mutation on the binding to RDM15 to chromatin.

Answer: We thank the reviewer's suggestion. We agree with the reviewer that ChIP-seq or ChIP-qPCR using wild type and mutated RDM15 will help to test the effect of the tudor mutation on the binding of RDM15 to chromatin. Currently, we only have older plants of *mutated RDM15-3xMYC/rdm15-3*, which is not suitable for ChIP experiment. However, to detect whether RDM15 binds to H3K4me1 in vivo and whether tudor domain is required for the binding of H3K4me1, we performed protein immunoprecipitation using *RDM15-3xFLAG/rdm15-3*, and *mutated RDM15-3xMYC/rdm15-3* plants with anti-FLAG antibody and anti-MYC, respectively, and then western blot with anti-H3K4me1 antibody was used to detect H3K4me1 (Supplementary Fig. 5c). The results showed that H3K4me1 was co-immunoprecipitated by RDM15-3xFLAG but not by mutated RDM15-3xMYC, suggesting that RDM15 is associated with H3K4me1 in vivo and mutations of tudor domain interrupt the binding of H3K4me1 .

Moreover, the H3K4me1 mark has been previously detected by ChIP-seq at whole-genome level in a previous study (Inagaki et al., EMBO J., 2017). The study showed that H3K4me1 is mainly located in genic regions. Because most of genic regions are not methylated and are

unlikely to be RdDM loci, it is necessary to determine the relationship between RDM15 ChIP-seq signals and H3K4me1 signals at the whole-genome level. The H3K4me1 ChIP-seq was reanalyzed in this study in order to compare the H3K4me1 signals and RdDM loci. However, it is necessary to analyze to what extent the RDM15 peaks overlap with H3K4me1 peaks at the whole-genome level and to determine whether RDM15 peaks overlap with H3K4me1 peaks at the genic regions that are not targeted by RdDM. These analyses are important for understanding whether RDM15 is specifically involved in RdDM or extensively associated with H3K4me1 at the whole-genome level. If RDM15 is enriched at all H3K4me1-enriched genic regions at the whole-genome level, RDM15 is unlikely to be a canonical component that is specifically required for RdDM.

Answer: We agree with the reviewer that the enrichment of RDM15 in genes suggested that RDM15 might not be a canonical component that is specifically required for RdDM. We discussed in the revised manuscript as “In addition, we found that RDM15 is also associated with H3K4me1 in genic regions, which are mostly not RdDM target regions. This result indicated that RDM15 may have RdDM-independent function in these regions.”